# Conversion and Obsessive–Phobic Symptoms Predict IL-33 and IL-28A Levels in Individuals Diagnosed with COVID-19

**DOI:** 10.3390/brainsci13091271

**Published:** 2023-08-31

**Authors:** Kristina Stoyanova, Drozdstoy Stoyanov, Steliyan Petrov, Alexandra Baldzhieva, Martina Bozhkova, Mariana Murdzheva, Teodora Kalfova, Hristina Andreeva, Hristo Taskov, Petar Vassilev, Angel Todev

**Affiliations:** 1Research Institute at Medical University of Plovdiv, Research Group “Translational and Computational Neuroscience”, SRIPD, Medical University of Plovdiv, 4002 Plovdiv, Bulgaria; drozdstoy.stoyanov@mu-plovdiv.bg; 2Department of Psychiatry and Medical Psychology, Medical University of Plovdiv, 4002 Plovdiv, Bulgaria; 3Department of Medical Microbiology and Immunology, Medical University of Plovdiv, 4002 Plovdiv, Bulgaria; steliyan.petrov@mu-plovdiv.bg (S.P.); alexandra.baldzhieva@mu-plovdiv.bg (A.B.); martina.bozhkova@mu-plovdiv.bg (M.B.); mmurdjeva@yahoo.com (M.M.); teodora.kalfova@mu-plovdiv.bg (T.K.); hristina.andreeva@mu-plovdiv.bg (H.A.); 4Laboratory of Clinical Immunology, University Hospital “St. George”, 4002 Plovdiv, Bulgaria; 5Research Institute at Medical University of Plovdiv, 4002 Plovdiv, Bulgaria; hristo.taskov@mu-plovdiv.bg; 6Department of Infectious Disease, Parasitology, and Tropical Medicine, Medical University of Plovdiv, 4002 Plovdiv, Bulgaria; petar.vasilev@phd.mu-plovdiv.bg (P.V.); angel.todev@mu-plovdiv.bg (A.T.)

**Keywords:** anxiety, depression, IL-28A and IL-33, SARS-CoV-2 infection

## Abstract

The first epidemiological wave of the incidence of COVID-19 in Bulgaria was registered in June 2020. After the wave peak, we conducted a study in persons diagnosed with COVID-19 (N = 52). They were assessed with the anxiety–depressive scale (ADS), including basic (BS), vegetative (VS), conversion (CS), obsessive–phobic (OPS), and depressive (DS) symptoms. ADS assessment of individuals diagnosed with SARS-CoV-2 indicated a correlation between OPS and IL-33 values. IL-10 levels were higher than reference ranges in all patients. Multiple linear regression analyses demonstrated that combination of CS and OPS explained 28% of IL-33 levels, while combination of symptoms from all ADS dimensions explained 24% of IL-33 levels. It was also found that 21% of IL-28A levels was explained from the combination by all ADS dimensions, whereas OPS was the predictor for lower concentrations. The obtained results revealed meaningful correlations between psycho neuro–immunological factors in pathogenesis of illness from the coronavirus infection.

## 1. Introduction

Globally, SARS-CoV-2 infection causes anxiety and despair due to its contagiousness, lethality, and lack of etiologic treatment [1]. In this early and critical period of the pandemic, the inflammatory cascade was described and observed in a large number of cases, including severe ones [2]. The pathogen was found to induce an intense immune and inflammatory response and rapidly release a large number of cytokines [3]. The tumor necrosis factor (TNF-α), IL-6, and IL-10 concentrations are the most important mediators of cytokine storm formation, and their levels clearly distinguish mild from severe cases of COVID-19 and are associated with poor prognosis of the disease. IL-6 is an anti- and pro-inflammatory cytokine whose expression is highly elevated in COVID-19 patients with lung infection [4]. IL-10 is an anti-inflammatory cytokine, also called cytokine synthesis inhibitory factor. IL-10 serum levels found in SARS-CoV-2 infection were dramatically elevated in intensive care unit patients [4]. Evidence from the past few years, points to an observable connection between higher serum levels of IL-10 and the presence of neurological symptoms [5]. During the clinical course of the infection, various neurological and psychiatric symptoms, such as confusion, disturbances in consciousness, loss of smell and taste, headache, seizures, paresis, problems with concentration, and changes in behavior were found. The risks of activation of the nervous system by SARS-CoV-2 are discussed in the following variants: through a direct effect of virus penetration, through a maladaptive immune response, through hypoxia in pneumonia leading to brain dysfunction, and in disorders of vascular endothelium and coagulation, leading to strokes [6].

Neuroinflammation is gaining increasing recognition in the pathogenesis of various neurological diseases associated with cognitive dysfunctions, such as Alzheimer’s disease, multiple sclerosis, ischemic stroke, traumatic brain injury, and central nervous system infections [7]. IL-33 has been found to have opposite effects in the progression of diseases leading to cognitive impairment. Moderate levels of IL-33 can limit nerve damage and slow disease progression by promoting the polarization of microglia and immunocompetent T-helper (Th) cells toward anti-inflammatory phenotypes and by enhancing the phagocytic activity of microglia to facilitate the clearance of dangerous substances and necrotic cells [7]. On the other hand, prolonged and persistent inflammatory responses can increase the production in IL-33 and other molecules by immune cells, leading to amplification loops in the brain and dysregulation of autophagy, apoptosis, and synaptic plasticity. These processes exacerbate neuronal damage, ultimately leading to cognitive impairment. The switch between protective and destructive effects in IL-33 concentration dynamics remains unclear [7], although there is evidence showing a correlation between disease severity and the increase in IL-33 levels in the serum [8]. Moreover, there is evidence for the role of IL-33 in neuroinflammatory processes in the major depressive disorder as well as in depression risk [9,10]. Elevated levels of IL-33, iNOS, HO-1, and MIP-1β concentrations have been reported in depressed individuals with and without the post-traumatic stress disorder (PTSD) [11]. The production of IL-33, as a result of SARS-CoV-2 infection, may be associated with the formation of pulmonary fibrosis by stimulating the production of various other cytokines. Interferon lambda-2 (IFN-λ2), or IL-28A, is a subclass of IFN-λ that is also involved in the coronavirus infection [4]. Besides being an independent predictor of the severity of COVID-19 [12], serum IL-28A has been reported to be significantly lower in people with insomnia co-morbid with depression compared to people who have chronic insomnia [13]. IL-28A, IL-28B, and IL-29 (also referred to as type III interferons) constitute a new subfamily within the IL-10 family. These cytokines have potent anti-viral and anti-tumor properties and promising therapeutic potential, but their immunoregulatory functions will still need investigation [14]. Little is known about the role of IL-28, but therapeutic use of interferon alpha has been suggested to be associated with a significant incidence of depression and suicidality [15]. It has been hypothesized that interferon therapy may induce depression [16].

Evidence has been accumulated to support the theory of immune dysregulation in elevated levels of inflammatory markers, such as C-reactive protein (CRP), IL-6, and TNF-α in depressed individuals. Although depression shows a high degree of co-morbidity with anxiety disorders, not all evidence for immune dysregulation in depression and co-morbid anxiety is consistent [17]. In general, anxiety disorder involves an excessive or inappropriate state of arousal characterized by foreboding, uncertainty, or fear. According to the severity and duration of symptoms and specific behavioral characteristics, anxiety disorders are classified into several general categories: generalized anxiety disorder, panic disorder, phobias, obsessive–compulsive disorder, and post-traumatic stress disorder. Clinical research has shown that impairments in quality of life and psychosocial functioning in generalized anxiety disorder are similar to those in major depressive disorder [18]. Symptoms of depression and anxiety have been found in patients who have recovered from COVID-19, in people who have been in contact with sick people, and during quarantine. The coronavirus was found to be associated with increased levels in affective (crucial in depression, anxiety, and melancholy) and physio–somatic symptoms of chronic fatigue, as well as cognitive symptoms and insomnia. Patients with COVID-19 are more likely to experience neuropsychiatric syndromes due to the stigma associated with the disease and anxiety about the impact of the infection. Mental and physical fatigue, loss of concentration, neurocognitive deficiency, headache, and myalgia are more common in them. Evidence has been generated showing that the effects of SARS-CoV-2 infection on neuropsychiatric symptoms are mediated by activated immunoinflammatory pathways. In COVID-19, a common infectious immunoinflammatory core underlies lung lesions’ low peripheral oxygen saturation (SpO2) and immune activation indicated by elevated plasma levels of IL-6, IL-10, CRP and sRAGE (soluble receptor for advanced glycation end products) and decreased levels in albumin and calcium. SARS-CoV-2 can infect the brain and cause neuroinflammation [19].

Cytokine profiles in the context of significant mental disorders such as depression and anxiety are under-studied. The proposed hypothesis is that there exists a relation between neurotic symptoms and target interleukins in SARS-CoV-2 infection that appears to also play a pivotal role in the neuro–inflammatory processes implicated in these disorders.

## 2. Materials and Methods

### 2.1. Participants and Procedure

After the peak of the first epidemiological wave of SARS-CoV-2 in Bulgaria, registered in June 2020 [20], the recruitment of persons diagnosed with COVID-19 for voluntary participation in our project research was initiated. The ADS questionnaire was designed to assess anxiety–depressive status in Bulgarian population in accordance with ICD-9 and ICD-10 [21]. The sample was compiled from the results of 52 persons who were asked to answer the questions in person and in writing within the time from confirmation of the COVID-19 infection and no later than the 20th day. The period of stay was different for each of them. Peripheral venous blood samples were collected from the patients on the 14th day post-hospitalization. Serum was isolated from blood samples and levels of IL-10, IL-28A, and IL-33 were evaluated in all individuals with COVID-19, as potential markers of anxiety-depression states were also recently identified in the literature in the SARS-CoV-2 cytokine profile [4,9,19]. 

Inclusion criteria for this study were set as follows: participants aged between 18 and 80 years; no confirmed autoimmune disorders or chronic inflammatory conditions; males and females are eligible for inclusion; no history of severe allergies or adverse reactions to immunomodulatory drugs. Participants were willing and able to provide informed consent for participation in this study. Exclusion criteria were defined as: subjects outside the age range from 18 to 80 years of age; diagnosed systemic autoimmune disorders (e.g., rheumatoid arthritis and/or lupus) or chronic inflammatory conditions (e.g., Crohn’s disease); pregnant or breastfeeding individuals due to potential effects on the immune system; currently intaking immunosuppressive medication that could interfere with the study’s objectives as well as with a history of severe allergic reactions; recent participation in immunomodulatory trials within the last six months—to avoid potential confounds. Furthermore, individuals unwilling or unable to follow the study’s protocol or attend required follow-up visits have been excluded.

### 2.2. Measures

ADS was originally designed as a neurotic–depressive test by Todor G. Tashev in 1969 [21] and is comprised of the following scales: Basic Syndrome (e.g., I feel that I am not good with nerves, I find it difficult to concentrate (I am distracted), my mood changes easily, I feel like my memory is fading); Vegetative Syndrome (e.g., I often have palpitations, I often get headaches, I get tired easily, I feel burning, pains and stabbings all over my body, I often have constipation or diarrhea, I don’t sleep well and have nightmares); Conversion Syndrome (e.g., I feel something like a ball in my throat, my legs and arms are tingling, I feel a trembling in my body, my jaws are tensed up); Obsessive–Phobic Syndrome (e.g., I keep feeling like something bad is going to happen to me, I am afraid of diseases, the same thoughts keep bothering me, I often think about death, everything seems unreal to me (like in a dream); and Depressive Syndrome (e.g., it is like I am estranged from myself, I cannot be joyful, I feel alienated from people and nature, I lack desire for everything, I often feel like crying, I am sad for no reason). Participants were instructed to assess the presence of the listed emotions and states in themselves during the past two weeks. Each positive answer carries 1 point. The clinical–diagnostic assessment is defined as follows: Practically healthy (up to 7 positive answers); pre-neurotic, Borderline state, or tendency to a neurotic state (from 8 to 17 positive answers); expressed neuroticism in three degrees– Mild neuroticism (18 to 22 symptoms), Average neuroticism (from 23 to 32 symptoms), and *severe neuroticism* (more than 32 neurotic symptoms present) [21]. The questionnaire contains two more scales— Etiological questions (e.g., I have problems at home, at work, I had a joyless childhood) and Control questions (e.g., sometimes I am envious, I am not always sincere) and contained a total of 65 items. Their ratings were intended to guide pre-processing of the data and serve as inclusion/exclusion criteria. 

A comprehensive review of the existing literature did not reveal any universally established reference values for IL-10, IL-28A, and IL-33. In that regard, and given the context of our study’s unique parameters, we embarked on an independent approach to derive relevant reference ranges. In our search for established and reliable reference values, we selected studies that aligned with recent advances and adhered to stringent inclusion criteria. A pivotal criterion was the inclusion of a robust control group, comprising a minimum of 25 individuals who were deemed healthy. This emphasis on a substantial control group ensured that the reference values we extracted were truly representative of a normative range within the broader population. Furthermore, the diversity in experimental techniques and methodologies, including the application of different assay kits for cytokine evaluation in serum, motivated an additional level of scrutiny. Understanding the intricacies associated with various assay kits, we selected studies that employed similar evaluation reagents to ours. This careful alignment was vital in mitigating the potential discrepancies introduced by disparate detection limits of various kits. By harmonizing the methodologies as closely as possible, we aimed to enhance the accuracy and relevance of the reference values we ultimately incorporated. As a result of this rigorous assessment, we acknowledge the following reference ranges as integral to our study’s context. For IL-10, a normal range of 0–5 pg/mL [22,23] was established based on our review of pertinent literature. Similarly, for IL-28A, we identified a normal range of below 25 ng/L [24]. Lastly, the reference range for IL-33 was determined to be <48.7 ng/L [25,26]. 

### 2.3. Statistical Analysis

Immunological assessment of cytokine levels was executed by enzyme-linked immunosorbent assay (ELISA), and the results were analyzed spectrophotometrically (BioTekTM). Main and routine statistical analyses were performed with SPSS Version 28.0. To compare ADS between patients by gender, independent samples *t*-test were performed. For assessment on the influence of neurotic symptoms on interleukins levels in patients, multiple linear regression analyses were performed as the prerequisites for this analysis were met. Results were considered statistically significant when *p* < 0.05.

## 3. Results

Distribution of the ADS data was normal, except for the kurtosis values on the CS and DS scales which were slightly above the range [−2, 2] (Appendix A). Internal consistency of items was excellent (Cronbach’s Alpha coefficient = 0.636–0.934) (Appendix A). The cohort was not representative of the population. The ratio of gender in participants was equal (26 men vs. 26 women) and the average age was 60.5 years (SD = 19.4).

The comparative analysis by gender in individuals with SARS-CoV-2 infection showed that the mean values of women on the vs. scale (M _women_ = 6.31, SD _women_ = 3.51, M _men_ = 4.20, SD _men_ = 2.96, *p* < 0.012) were significantly higher than those of men (Appendix A). 

Results showed a tendency of borderline state with the highest percentage (18.75%) being in the female patients. In the male group, the vast majority (25%) were classified as practically healthy. Less than 5% of the COVID-19 patients were categorized as having severe neuroticism (Appendix A).

The ADS by age groups in the patients was analyzed in more detail, since an immunological assessment was available for them during their hospitalization. It was found that in the group’s level borderline state, mild, average and severe neuroticism seemed to have a prevalence in the groups aged 41–60 years and 61–80 years in a small number of people. In patients aged above 81 years, no severe states of neuroticism were observed and most of them (N = 8) were practically healthy (Appendix A). 

Our results depict that the IL-10 levels ($\bar{x}$ = 339.68 ± 22.38 SD) in all patients were elevated in comparison to reference values (Appendix A). After evaluating the cytokine serum levels, it was discovered that in the group aged 61–80 years contained most patients (34%) with IL-10 values above the reference range (Table 1, Appendix A). 

The levels of IL-28A ($\bar{x}$ = 61.40 ± 13.04 SD) were abnormally high in 29 individuals. It was found that the group in which most patients (N = 8) had their cytokine values in the reference range was the group containing individuals <40 years of age. On the other hand, the highest number of patients (N = 13) whose IL-28A values were above the reference range, were in the group aged 61–80 years, followed by the age group of patients above 81 years of age (N = 7) (Table 1, Appendix A).

Distribution analysis of serum IL-33 levels ($\bar{x}$ = 165.05 ± 35.02 SD) found that most patients (N = 16) who had concentrations within the reference range, were those belonging to the group aged 61–80 years. Regarding the values above the reference range, it was discovered that most patients constituted from the age group between 41 and 60 years of age (N = 10) (Table 1, Appendix A).

A significant positive correlation was found only between IL-33 levels and the OPS scale (r = 0.289, *p* = 0.046) in persons diagnosed with COVID-19 (Appendix A).

In the immuno–demographic perspective, IL-10 seems to have the strongest associations with IL-28A, IL-33 and all age groups, except for the oldest group. At the same time, high values of IL-28A are most significantly correlated with the age range between 61 and 80 years of age, and high values of IL-33 are correlated with the age range between 41 and 60 years of age (Figure 1).

In the perspective of psycho–immunological assessment and gender (in which we have excluded IL-10), the reference values of IL-33 seem to have the strongest relation to men, who in turn correspond to the category of practically healthy in this sample. Women were associated with borderline state, as well as and with reference and high values of IL-28A and IL-33. Perhaps the observation that the reference ranges of IL-28A were equally associated with both sexes and borderline state was relevant to our hypothesis (Figure 2). 

Results of the multiple linear regression analysis by enter method for the joint influence of ADS symptoms on cytokine levels are shown in Table 2. In the first prediction model, the combination of symptoms of all syndromes had no significant influence on the variation of IL-10. In the second prediction model, 21% of the variation in IL-28A was significantly explained by symptoms of all syndromes. Case-wise diagnostics identified data for anomalous observations in two individuals. They were removed as outliers. This result was reported on the data from 50 persons. In the next prediction model, IL-28A values from CS and OPS symptoms were tested. This combination explained 10% of the variation in IL-28A (in 50 persons). OPS was a negative predictor of IL-28A values in any test. In the fourth prediction model, symptoms of all syndromes explained 24% of the variation in IL-33 (in 51 persons). The data of one person was removed (due to being an outlier) in case-wise diagnostics. In the last fifth prediction model, 28% of the variation in IL-33 was explained by symptoms of CS and OFS (in 51 persons).

## 4. Discussion

The ADS assessment in persons diagnosed with COVID-19 in this study had a contextual motivation—a subsiding first wave of an incipient pandemic. We distributed efforts in two focuses adequate to the objectively stressful situation. On the one hand, to measure anxiety–depressive states with a short and effective method. On the other hand, to relate ADS to the immune status in a cohort of persons diagnosed with SARS-CoV-2 infection. Based on the well-established cytokine profile in COVID-19 [4], and after a thorough literature review for cytokines that regulate immune and inflammatory responses in psychological disorders in the anxiety–depressive spectrum [9,11,12,14,15,16], a psychoneuro–immunological research framework was focused on. IL-10, IL-28A, and IL-33 were identified as targetable indicators of neuroimmune dysregulation associated with the coronavirus disease.

In patients, a tendency towards the borderline state of ADS was observed. Women reported significantly higher mean values on the vs. scale than men. The highest percentage among female patients was classified in the borderline state, while the majority of male patients were categorized as practically healthy. Furthermore, the prevalence of anxiety–depressive states was prominent in the age groups between 41 and 60 years of age as well as between 61 and 80 years of age.

The cytokine profile of the inflammatory cascade in SARS-CoV-2 was well defined, with concentrations of TNF-α, IL-6 and IL-10 distinctly mediating the cytokine storm and clearly differentiating mild from severe cases of COVID-19 [4]. Our analysis revealed elevated cytokine levels of IL-10 in all patients diagnosed with COVID-19. These findings are consistent with previous studies demonstrating increased IL-10 levels in COVID-19 patients, indicating an immune response modulation [27]. Notably, the age group between 61 and 80 years of age exhibited the highest number of patients with IL-10 values above the reference range, suggesting a potential association between age-related immune dysregulation and cytokine expression. IL-10 is an anti-inflammatory cytokine that plays a crucial role in regulating immune responses and controlling inflammation. While IL-10 is primarily recognized for its immunosuppressive properties, it has also been implicated in neuro–inflammation and neuro–degenerative diseases [28]. Other research teams have reported that a significant increase in IL-10 and TNF-α is observed in more severe cases of patients with COVID-19 [29,30,31]. Furthermore, an interesting finding was reported by Azaiz et al. in 2022, who found that the IL-6/IL-10 ratio can be used as a very powerful predictor of disease severity [32].

In terms of IL-28A levels, abnormally high values in a significant proportion of patients, particularly in the group aged 61–80 years, were found. This age group demonstrated the highest number of patients with IL-28A levels above the reference range. Conversely, patients below 40 years of age had a higher proportion of IL-28A values within the reference range. These findings indicate a potential age-related impact on IL-28A expression, possibly implicating its role in immune responses and disease progression among older individuals. Due to the limited data available concerning the role of IL-28A in COVID-19, one can only hypothesize about its function. IL-28A, also known as interferon lambda 2 (IFN-λ2), is a type III interferon along with IL-28B and IL-29, and they all play a role in antiviral immune responses. While the primary focus of IL-28A has been its antiviral activity, recent studies have also suggested its involvement in neuro–inflammatory processes. The role of IL-28A might be important regarding the disease’s severity, as some research teams report. Fukuda et al. discovered that serum levels of IL-28A tend to be lower in severe cases, proposing that its role might be involved in the prevention of severe disease by promoting viral shedding [12].

The IL-33 level investigation revealed that the majority of patients aged 61–80 years had concentrations within the reference range, indicating a relative stability of IL-33 expression in this age group. However, a non-significant number of patients in all age groups had IL-33 levels above the standard range. Similar results were observed by Majeed et al. in their study [33]. IL-33, a member of the IL-1 family, is an alarmin cytokine with crucial roles in tissue homeostasis and repair, type 2 immunity, allergic and non-allergic inflammation, viral infection, and cancer [34]. In terms of the COVID-19 infection, scientists have proved that higher levels of IL-33 are associated with disease severity and progression [8]. Other researchers that focused on cytokine roles in neuroinflammation suggest that IL-33 may at least in part be responsible for inducing a transition from the Th1 to the Th2 immune response and for inhibiting the Th17 immune response, which can act as an anti-inflammatory and lessen cognitive impairment following CNS damage, such as in severe cases of COVID-19 [7,35]. 

What IL-28A and IL-33 have in common (more specifically, lower concentrations of IL-28A and high concentrations of IL-33) is their expression in neuroinflammatory processes, depression, with and without PTSD, risk of depression as well as in insomnia comorbid with depression [9,11,15,16]. SARS-CoV-2 in turn was associated with increased levels in affective and physio–somatic symptoms of chronic fatigue, cognitive symptoms, insomnia, and neuro–inflammation [19]. In our patients’ sample, significant correlation was found only between the OPS scale and IL-33 values. However, regarding this group, the visualization of the variables generated relationships maps that were consistent with the literature data and our hypothesis. The following relationships were observed. First, the strongest associations were seen between IL-33 reference ranges and men, who in turn were classified as practically healthy. Second, the strongest associations between reference and high ranges of IL-28A and of IL-33 and women, were classified in the borderline state.

Multiple regression analysis confirmed significant relationships between IL-28A, IL-33, and neurotic symptoms in the anxiety–depressive spectrum. Our most valuable finding was that the combination of anxiety–depressive symptoms explained 21% of the variation in IL-28A and 24% of the variation in IL-33. Specifically, the combination of CS and OPS explained 10% of the variation in IL-28A and 28% of the variation in IL-33, and the OPS was the predictor for lower IL-28A concentrations. It should be noted that in the two age groups with the highest neuroticism, patients aged 41–60 years had the highest concentrations of IL-33 and expression of IL-28A in the reference ranges, whereas the IL-28A values were abnormally high in individuals aged 61–80 years. Predominantly, those were women with conversion and/or obsessive–phobic symptoms. The results support the initial hypothesis and recent studies, reporting associations between neurotic symptoms and IL-28A and IL-33.

## 5. Limitations 

It should be noted that our sample was relatively small to draw a clear conclusion. Cytokine interactions, as implicated from the immune network theory of Niels Jerne [36,37], are far more complex and demand further comprehensive investigations in terms of their relations to abnormal mental conditions. The sample was also missing a control group. However, the study design aimed at investigating disturbances in a specific disease population; therefore, recruiting a healthy control population was not required.

Results need to be replicated in a larger sample size with extended cytokine profiling. 

Longitudinal studies that follow individuals over time and investigate the temporal dynamics between symptoms and cytokine levels may provide further insights into this complex association.

## 6. Conclusions

There has been a reported association between the anxiety–depressive spectrum phenomena and IL-28A and IL-33 levels in individuals diagnosed with COVID-19. A thorough explanatory model for the mechanisms of interference between these cytokines and pathomorphosis of anxiety–depressive disorders require profiling of a wider set of serum biomarkers and reliable psychodiagnostic assessment tools. However, our findings contribute to the understanding of the psycho–somatic interactions implicated in the COVID-19 infection. Fundamentally, the presence of obsessive–phobic and conversion symptoms predict altered cytokine functions that take active part in the pathogenesis of the COVID-19 infection and determine its clinical course. The mutual conditioning between OPS, CS, ADS, and corresponding modulation of cytokine levels is not completely understood as it is part of a complex immune interactions network [36,37]. Still, there is evidence suggesting a potential association between certain cytokines and obsessive–phobic symptoms [38,39]. Cytokines are involved in various physiological processes, including immune response modulation and inflammation regulation. Some studies have found associations between cytokine levels and psychiatric symptoms, but establishing causal inferences is still challenging. 

## Figures and Tables

**Figure 1 brainsci-13-01271-f001:**
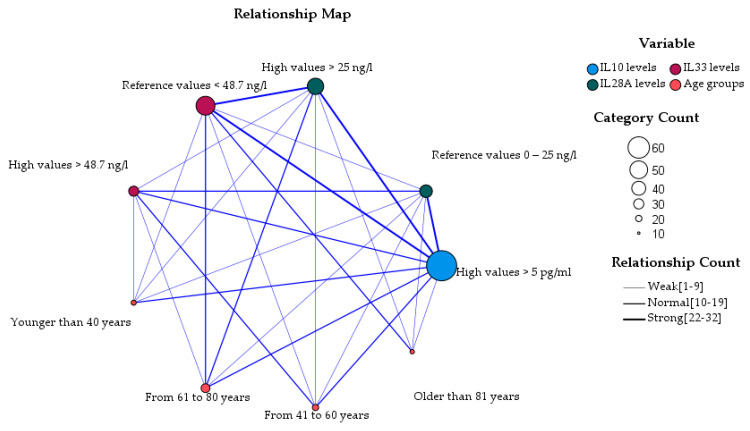
Relationship between interleukins IL-10, IL-28A and IL-33 ranges, and age groups in persons diagnosed with COVID-19 (N = 52).

**Figure 2 brainsci-13-01271-f002:**
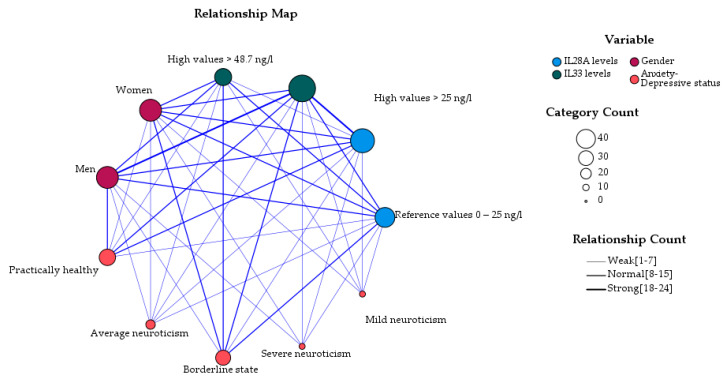
Relationship between interleukins IL-28A and IL-33 ranges, age groups, and ADS in persons diagnosed with COVID-19 (N = 52).

**Table 1 brainsci-13-01271-t001:** Interleukin reference ranges in persons diagnosed with COVID-19 14 days after confirmed infection (N = 52).

Reference Ranges	IL-10 (pg/mL)	IL-28A (ng/L)	IL-33 (ng/L)
	Men–Women (%)	Men–Women (%)	Men–Women (%)
<0 pg/mL	-	-	-
0–5 pg/mL	-	-	-
>5 pg/mL	26–26 (50%–50%)	-	-
<0 ng/L	-	-	-
0 ng/L–25 ng/L	-	12–11 (52.2%–47.8%)	-
>25 ng/L	-	14–15 (48.3%–51.7%)	-
<48.7 ng/L	-	-	18–15 (54.6%–45.4%)
>48.7 ng/L	-	-	8–11 (42.1%–57.9%)

**Table 2 brainsci-13-01271-t002:** Multiple linear regression analysis by enter method to predict interleukins ranges in persons diagnosed with COVID-19 by ADS.

Predictors	R^2^	R	B	β	t	p
Variables Entered ^1^	0.123	0.467				0.064
BS			12.097	0.192	0.859	0.396
VS			16.553	0.351	1.676	0.101
CS			51.435	0.434	1.732	0.091
OPS			−51.959	−1.016	−3.005	0.005
DS			10.039	0.136	0.579	0.566
Variables Entered ^2^	0.212	0.548				0.011
BS			−1.980	−0.072	−0.337	0.738
VS			3.938	0.193	0.968	0.339
CS			41.392	0.811	3.416	0.001
OPS			−17.961	−0.816	−2.536	0.015
DS			−6.010	−0.189	−0.844	0.404
Variables Entered ^3^	0.095	0.366				0.040
CS			33.050	0.474	1.996	0.052
OPS			−18.748	−0.623	−2.625	0.012
Variables Entered ^4^	0.241	0.569				0.005
BS			3.470	0.038	0.183	0.856
VS			−2.839	−0.043	−0.208	0.836
CS			−142.984	−0.859	−3.630	<0.001
OPS			62.146	0.868	2.749	0.009
DS			15.675	0.151	0.669	0.507
Variables Entered ^5^	0.283	0.561				<0.001
CS			−136.801	−0.822	−3.853	<0.001
OPS			68.318	0.954	4.476	<0.001

Predictors: BS—basic syndrome, VS—vegetative syndrome, CS—conversion syndrome, OFS—obsessive–phobic syndrome, DS—depressive syndrome. ^1^ Dependent variable: IL-10. ^2^ Dependent variable: IL-28A. ^3^ Dependent variable: IL-28A. ^4^ Dependent variable: IL-33. ^5^ Dependent variable: IL-33.

## Data Availability

The data presented in this study are available on request from the corresponding author.

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
