# Peer review of "Conversion and Obsessive–Phobic Symptoms Predict IL-33 and IL-28A Levels in Individuals Diagnosed with COVID-19"

_brainsci, 2023, doi:10.3390/brainsci13091271_

Round 1

Reviewer 1 Report

Title: Conversion and obsessive-phobic symptoms predict IL-33 and IL-28A levels in individuals diagnosed with COVID-19

The study conducted in Bulgaria during the first wave of COVID-19, focusing on the relationship between psychological symptoms, immunological markers, and the course of the illness. The study's findings suggest that there may be connections between psychological factors and the immune response in individuals affected by COVID-19. This study contributes to the understanding of the complex interactions between mental health and immune function during the course of the disease.

Introduction

Writing is well-structured and generally clear in conveying the information from the provided introduction. However, there are a few areas where minor improvements can be made for clarity and coherence.

Materials and Methods

Ethical Approval Statement: Don't forget to include a statement indicating that ethical approval was obtained for the study. This is a standard requirement in research involving human participants.

Regarding the reference values, to enhance transparency, the author could briefly describe the sources of the mean values used for reference ranges.

Results

Abbreviation under Table should have full statement of all parameters (predictors).

Discussions and conclusions

There is a continuation or an elaboration of the abstract, focusing on the relationship between interleukins (IL-28A and IL-33) and neurotic symptoms in COVID-19 patients. Overall, the writing maintains a consistent scientific tone and conveys the research findings clearly. However, there are some areas where the text can be improved for clarity and readability.

Reviewer 2 Report

A body of evidence has emerged to explain the role of immune deregulation in individuals with SARS-CoV-2 infection. The notable co-occurrence of depression and anxiety disorders in these patients revealed the importance of exploring the link between deregulation of the immune system and the presence of neurotic symptoms in these patients. In this work, authors hypothesized a potential correlation between neurotic symptoms and specific interleukins in SARS-CoV-2 infection.

I have some questions for the authors, and also some language suggestions.

1.     Could you provide more data for the physicians group related to their exposure to the SARS-CoV-2 infection and the potential disease before entering the study and measuring cytokines? How was this group chosen and what factors were considered in its selection?

2.     In the study, the authors mention that the reference ranges for analyzed cytokines are not established. Could you provide insight into what these reference ranges were based on and how you determined them?

3.     Do you have any data regarding potential confounding factors, such as comorbidities that might influence the relationship between psychological symptoms and immunological data in both groups analyzed?

4.     In the Conclusion section, the authors should, with a couple of sentences, elaborate on how their findings contribute to the understanding of the psychoneuroimmunological factors involved in the pathogenesis of coronavirus infection.

5.     Given the relatively small sample sizes for both groups, were any statistical measures taken to address potential limitations arising from the sample size?

I also have some language suggestion for the authors:

1.     In the Abstract, please, reformulate the sentence (line 23): In both of these groups, we have assessed….

2.     Line 26 – showed a correlation with IL-10, IL-28, and IL-33 levels.

3.     Line 28 – in IL-33 levels.

4.     Please, reformulate the last two sentences in line 29 from Abstract and make them clearer.

5.     Please, provide references in lines 46, 61, 66.

6.     Line 103: Please,  use the term crucial instead of key.

7.  Line 112 - sRAGE - Soluble Receptor for Advanced Glycation End Products, add term end.

8.     Lines 115-119: Please, reformulate and make two sentences.

9.     Line 280: Please, use the term focused instead of consolidated.

10.  Lines 370 and 378 : Please, clarify and reformulate sentences.

11.  There are a lot of comma signs missing in the text. Please, correct this.

The text is missing a lot of commas. I suggested corrections for certain sentences to the authors.

Reviewer 3 Report

Dear Authors, although the concept of your study is interesting the methodology is weak. The patients’ sample size is limited and potentially suffers a selection bias. Results based upon 30 COVID-19 patients being in Borderline, Mild, Average and Severe neuroticism states and biological markers (interleukins) are highly ambiguous. It is not clear why the researchers recruited the 55 medical specialists if they cannot serve as control group (Just to estimate the internal consistency of the measurement tool or to indicate the impact of the pandemic on medical professionals, which is not the purpose of the study?) In studies with biological markers control groups are necessary.

In introduction section lines 110-114 refer to the clinical course during the infection or post-COVID?

It is not clear if the COVID-19 patients completed the questionnaire at the same day they gave blood samples. Also, other important information is missing such as past psychiatric history; concomitant immune modifying medications etc. and exclusion criteria need to be enriched.

In lines 172-173 authors state: ‘To more in-depth assessment the influence of neurotic symptoms on interleukins levels in patients, we performed Multiple linear regression analysis’. Actually, interleukin levels as independent factors influence neurotic symptoms, not the other way around. In this sense, even the title needs rephrasing:  IL-33 and IL-28A levels predict conversion and obsessive-phobic symptoms   in individuals diagnosed with COVID-19.  

Please verify sample adequacy (using the G-Power) for your regression models.

Stating that multiple linear regression models satisfy key assumptions is not enough. You need to provide specifics. 

In lines 275-276 you mention that the objective was: ‘to measure anxiety-depressive states with a short and effective method’. There are other brief and popular rating scales measuring anxiety and depression in hospitalized patients. The scale you have chosen is extensive and the neurotic-depression concept is a bit obsolete.  

Extensive editing of English language is required. 

Round 2

Reviewer 3 Report

Dear Authors,

The article has substantially been improved. No further notifications.